# Serological Biomarkers of Extracellular Matrix Turnover and Neutrophil Activity Are Associated with Long-Term Use of Vedolizumab in Patients with Crohn’s Disease

**DOI:** 10.3390/ijms23158137

**Published:** 2022-07-23

**Authors:** Marta S. Alexdottir, Arno R. Bourgonje, Morten A. Karsdal, Martin Pehrsson, Roberta Loveikyte, Hendrik M. van Dullemen, Marijn C. Visschedijk, Eleonora A. M. Festen, Rinse K. Weersma, Klaas Nico Faber, Gerard Dijkstra, Joachim H. Mortensen

**Affiliations:** 1Biomarkers and Research, Nordic Bioscience, 2730 Herlev, Denmark; mk@nordicbio.com (M.A.K.); mpe@nordicbio.com (M.P.); jhm@nordicbio.com (J.H.M.); 2Department of Gastroenterology and Hepatology, University Medical Center Groningen, University of Groningen, 9713 GZ Groningen, The Netherlands; a.r.bourgonje@umcg.nl (A.R.B.); r.loveikyte@umcg.nl (R.L.); h.m.dullemen@umcg.nl (H.M.v.D.); m.c.visschedijk@umcg.nl (M.C.V.); e.a.m.festen@umcg.nl (E.A.M.F.); r.k.weersma@umcg.nl (R.K.W.); k.n.faber@umcg.nl (K.N.F.); gerard.dijkstra@umcg.nl (G.D.)

**Keywords:** inflammatory bowel disease, biomarkers, extracellular matrix, neutrophils, vedolizumab

## Abstract

Crohn’s disease (CD) is a relapsing-remitting inflammatory disease of the gastrointestinal (GI) tract characterized by increased extracellular matrix (ECM) remodeling. The introduction of the α4β7-integrin inhibitor vedolizumab (VEDO) has improved disease management, although there is a high rate of primary non-response in patients with CD. We studied whether ECM biomarkers of neutrophil activity and mucosal damage could predict long-term response to VEDO in patients with CD. Serum levels of human neutrophil elastase (HNE)-derived fragments of calprotectin (CPa9-HNE), and matrix metalloproteinase (MMP)-derived fragments of type I (C1M), III (C3M), IV (C4M), and VI (C6Ma3) collagen, type III collagen formation (PRO-C3), basement membrane turnover (PRO-C4) and T-cell activity (C4G), were measured using protein fingerprint assays in patients with CD (*n* = 32) before VEDO therapy. Long-term response was defined as VEDO treatment of at least 12 months. CPa9-HNE was significantly increased at baseline in non-responders compared with responders (*p* < 0.05). C1M, C3M, C4M, C6Ma3, and PRO-C4 were also significantly increased at baseline in non-responders compared with responders (all *p* < 0.05). All biomarkers were associated with response to VEDO (all *p* < 0.05). To conclude, baseline levels of serum biomarkers for neutrophil activity and mucosal damage are linked to the pathology of CD, and are associated with long-term use of VEDO in patients with CD. Therefore, these biomarkers warrant further validation and could aid in therapeutic decision-making concerning vedolizumab therapy.

## 1. Introduction

Crohn’s disease (CD) is a chronic inflammatory disease of the gastrointestinal (GI) tract characterized by flares of intestinal inflammation that, if left insufficiently treated, may give rise to complications such as abscesses, fistulas, and strictures, often necessitating surgical interventions. The pathogenesis of CD is thought to be associated with genetic susceptibility, environmental triggers (e.g., lifestyle, dietary factors), and an abnormal immunological response to the gut microbiota [1,2]. Even though there is no cure for CD, disease management has substantially improved with the emergence of anti-tumor necrosis factor (TNF) antibodies, which are routinely used as first-line biologic treatment [3]. However, insufficient response to anti-TNF treatment has been reported to be up to 40% [4]. The α4β7-integrin inhibitor vedolizumab (VEDO) is a safe and effective alternative often used as a second- or third-line treatment option for patients experiencing anti-TNF failure. As studies have shown VEDO to be more effective at inducing response amongst biologic-naïve patients, biologic-exposed patients are more challenging to treat [5,6]. It is evident that there is an unmet need for monitoring and prediction tools capable of estimating the efficacy of a suggested treatment prior to induction—especially considering that the probability of success drastically declines with each line of treatment [7].

The extracellular matrix (ECM) is a crucial component of the intestinal tissue. It comprises a complex network of structural proteins such as collagens and is necessary for tissue function, structure, and homeostasis. Within the GI tract, endothelial- and epithelial cells reside on a specialized matrix called the basement membrane (BM), consisting mainly of type IV collagen synthesized by mesenchymal and enteroendocrine cells. Below the BM lies the interstitial matrix (IM), providing structural and functional integrity to the intestinal wall. It is rich in type I and III collagens, both of which are fibrillar collagens produced mainly by fibroblasts. In addition, type VI collagen can be found between the BM and IM, separating the two layers [8,9,10]. Due to the chronic inflammation present in CD, the ECM of the GI tract is constantly being remodeled, leading to a disruption in tissue homeostasis. This imbalance initiates a pro-inflammatory loop resulting in proteolytic cleavage of ECM components mediated by tissue-resident cells as well as migrating immune cells. These cells secrete proteases such as matrix metalloproteinases (MMPs), granzyme-B (GrzB), and human neutrophil elastase (HNE), which have been associated with the intestinal pathology observed in CD [11,12,13]. Eventually, these protease-derived ECM fragments are released into the circulation, where they can be detected and used as serological biomarkers reflecting ECM integrity and intestinal inflammation. To date, this has been demonstrated in several studies. For example, two previous studies showed that MMP-mediated degradation fragments of type I, III, and IV collagen are associated with active inflammation in patients with CD [14,15]. Similarly, a fragment of MMP-mediated type VI collagen degradation was elevated in serum from patients with GI disorders [16]. Further studies are required to validate the potential of these biomarkers as predictors of response to different biologics.

This study aimed to investigate whether biomarkers of ECM formation and degradation could identify and predict long-term response to VEDO treatment in patients with CD. Furthermore, we intended to assess the relationship between different levels of ECM degradation and treatment response.

## 2. Results

### 2.1. Cohort Characteristics

A total of 32 patients with CD were included in the study. Baseline demographic and clinical characteristics are shown in Table 1, stratified by long-term use (defined as >12 months) of VEDO treatment. The median age was 41 (35–60) and 42 (35–60) years for long-term users vs. discontinued users (*p* > 0.999), respectively. Patients who continued VEDO treatment for at least 12 months had significantly lower CRP levels at baseline compared with those who discontinued treatment: 2 (1–6) mg/L vs. 14 (8–26) mg/L (*p* < 0.001). No other significant differences were observed between both groups for the remaining clinical parameters. However, a noteworthy trend was observed for prior use of TNF-α-antagonists where fewer patients with long-term use of VEDO were anti-TNF experienced (88%) compared with patients who discontinued the use of VEDO, who were all (100%) previously exposed to TNF-α-antagonists. All patients were completely VEDO-naïve at the start of induction. Moreover, no differences were observed for subcategories of the Montreal disease classification (age, location, and disease behavior) between the two groups. Correlations were analyzed to investigate the relationship between measured biomarkers and disease activity (Appendix A). No significant correlations were found between the biomarkers and disease activity. However, a trend was observed where C4G showed a positive correlation with HBI (ρ = 0.35, *p* = 0.066). C3M showed a trend toward being positively correlated with the SES-CD score (ρ = 0.38, *p* = 0.097). Type III collagen formation (PRO-C3) trended towards being negatively correlated with having an active disease at baseline (PRO-C3: ρ = −0.34, *p* = 0.081).

### 2.2. Fragments of Type I, III, IV, and VI Collagen and Serum Calprotectin Are Elevated at Baseline in Patients with Crohn’s Disease Who Discontinue the Use of Vedolizumab within 12 Months after the Start of Induction

Differences in baseline serum levels of collagen formation and degradation, as well as human neutrophil elastase (HNE)-mediated calprotectin degradation, between patients who continued the use of VEDO for at least 12 months after the start of therapy and those who discontinued, are shown in Table 2. Fragments of type I (C1M), III (C3M), IV (C4M) and VI (C6Ma3) collagen degradation were elevated at baseline in patients who discontinued use of VEDO compared with long-term users (C1M: 108.6 [57.47–148.45] ng/mL vs. 36.1 [25.02–49.80] ng/mL, *p* = 0.001; C3M: 15.2 [13.07–16.44] ng/mL vs. 10.5 [9.43–11.81] ng/mL, *p* = 0.006; C4M: 36.5 [28.24–46.68] ng/mL vs. 24.5 [22.36–28.16] ng/mL, *p* = 0.010; C6Ma3: 0.8 [0.68–1.02] ng/mL vs. 0.6 [0.53–0.84] ng/mL, *p* = 0.015) (Table 2, Figure 1). Moreover, basement membrane turnover (PRO-C4), the ratio between type III collagen formation and degradation (C3M/PRO-C3), and two distinct type IV collagen turnover ratios (C4M/C4G, PRO-C4/C4G) were significantly higher at baseline in patients who discontinued treatment, compared with patients who continued treatment for at least 12 months (PRO-C4: 266.8 [207.20, 308.26] ng/mL vs. 182.0 [170.48, 203.68] ng/mL, *p* = 0.010; C3M/PRO-C3: 2.6 [1.98–3.13] vs. 1.6 [1.42–1.80], *p* = 0.008; C4M/C4G: 2.0 [1.40–2.44] vs. 1.4 [0.81–1.77], *p* = 0.020; PRO-C4/C4G: 14.5 [9.66–17.12] vs. 8.4 [6.02–12.56], *p* = 0.033) (Table 2, Figure 1). Additionally, patients who discontinued treatment had elevated baseline levels of the neutrophil activity marker CPa9-HNE compared with long-term users (401.2 [281.36–452.00] ng/mL vs. 231.7 [195.32–302.60] ng/mL, *p* = 0.003) (Table 2, Figure 1). The ratio between CPa9-HNE and T-cell mediated type IV collagen degradation (C4G) showed a trend towards elevated baseline levels in patients with discontinued use of VEDO compared with long-term users (18.5 [11.14–29.28] vs. 11.6 [6.46–16.43], *p* > 0.05). A trend was also observed for levels of type III collagen formation (PRO-C3), where patients with continued use of VEDO had higher levels at baseline compared with patients who discontinued treatment (6.4 [4.43–8.40] vs. 5.7 [5.34–6.49], *p* > 0.05) (Table 2). In summary, these results show that both neutrophil activity and ECM turnover were overall higher at baseline in patients who discontinued VEDO therapy within the first 12 months compared with patients who became long-term VEDO users.

Biomarker levels were compared between patients stratified by primary response (*n* = 10), secondary response (*n* = 7), or no response (*n* = 15) (as evaluated at 14 weeks after treatment initiation) to further characterize the biomarker profile of patients who continued VEDO treatment for at least 12 months (Appendix A). Baseline levels of C1M and CPa9-HNE were significantly lower in both primary responders (PR) and secondary responders (SR) compared with non-responders (NR) (C1M: 33.7 [26.46–49.21] ng/mL and 38.3 [25.38–58.60] ng/mL vs. 108.6 [57.47–148.45] ng/mL, *p* < 0.001; CPa9-HNE: 235.5 [202.61–287.24] ng/mL and 227.8 [181.74–280.28] ng/mL vs. 401.2 [281.36–452.00] ng/mL, *p* < 0.05 for both) (Figure 2, Appendix A). When compared with NR, the biomarkers C3M and C4M were significantly lower only in SR (C3M: 8.9 [7.21–11.20] ng/mL vs. 15.2 [13.07–16.44] ng/mL, *p* = 0.006; C4M: 22.5 [21.08–28.03] ng/mL vs. 36.5 [28.24–46.68] ng/mL, *p* < 0.05). The type IV collagen turnover ratio C4M/C4G was only significantly lower in PR at baseline when compared with NR (22.5 [21.08–28.03] vs. 36.5 [28.24–46.68], *p* < 0.05) (Figure 2, Appendix A). No significant differences were observed between PR and SR for the remaining biomarkers. However, relevant trends were noted for the biomarkers C3M, PRO-C3, C4M and C4G, where numerically higher levels were observed within PR compared with SR (C1M: 10.7 [10.02–14.68] ng/mL vs. 8.9 [7.21–11.20] ng/mL; PRO-C3: 7.4 [6.19–8.95] ng/mL vs. 4.9 [4.06–6.74] ng/mL; C4M: 24.9 [23.42–36.52] ng/mL vs. 22.5 [21.08–28.03] ng/mL; C4G: 24.1 [15.85–33.60] ng/mL vs. 19.2 [14.46–21.51] ng/mL, *p* > 0.05 for all) (Appendix A).

### 2.3. Discriminative Accuracy of Serum Levels of Type I, III, IV, and VI Collagen Fragments Regarding Long-Term VEDO Treatment

The discriminative capacity of the baseline biomarker levels regarding long-term use of VEDO was assessed using ROC statistics. Serum C1M levels, reflecting type I collagen degradation, and the neutrophil activity marker CPa9-HNE showed the best discriminative ability regarding long-term continuation vs. discontinuation of VEDO treatment (AUC with 95% CI: C1M 0.85 [0.75–0.98]; CPa9-HNE 0.81 [0.66–0.96], *p* < 0.001 for both). Similarly, baseline levels of degradation markers of type III (C3M), IV (C4M) and VI (C6Ma3) collagen, as well as turnover ratios of type III (C3M/PRO-C3) and IV collagen (C4M/C4G, PRO-C4/C4G), were able to discriminate between long-term continuation of VEDO treatment (AUC with 95% CI: C3M 0.79 [0.62–0.95], *p* = 0.001; C4M 0.77 [0.60–0.93], *p* = 0.002; C6Ma3 0.75 [0.58–0.92], *p* = 0.004; C3M/PRO-C3 0.78 [0.60–0.95], *p* = 0.002; C4M/C4G 0.74 [0.56–0.92], *p* = 0.009; PRO-C4/C4G 0.72 [0.54–0.90], *p* = 0.015). As an exploratory analysis, biomarkers were combined using multivariable logistic regression analysis into pairs to evaluate their combined discriminative value. The optimal combination of biomarkers associated with long-term use of VEDO constituted the combination of PRO-C4 and C6Ma3 with a corresponding AUC of 0.84 [0.70–0.98] (*p* < 0.001). The combinations of [C3M, C6Ma3] and [PRO-C4, C3M/PRO-C3] also demonstrated accurate discriminative ability between long-term VEDO users and non-users (AUC with 95% CI: C3M, C6Ma3 0.83 [0.69–0.97]; PRO-C4, C3M/PRO-C3 0.81 [0.65–0.97]; *p* < 0.001 for both) (Table 3).

Based on the optimal cut-off values determined for each biomarker, odds ratios were calculated accordingly (Table 4). The odds of treatment discontinuation were significantly increased for patients with biomarker levels above the given cut-off value for type I (C1M), III (C3M), IV (C4M), and VI (C6Ma3) collagen degradation as well as for the neutrophil activity marker CPa9-HNE, compared with patients that had levels below the cut-off value (OR [95% CI]: C1M 14.08 [2.35–59.09] *p* < 0.01; C3M 15.60 [2.79–80.36] *p* < 0.01; C4M 8.93 [1.62–42.45] *p* < 0.05; C6Ma3 15.75 [2.01–182.70] *p* < 0.01; CPa9-HNE 19.43 [2.36–255.20] *p* < 0.01). Biomarker ratios of neutrophil/T-cell activity (CPa9-HNE/C4G), type III (C3M/PRO-C3) and type IV collagen (C4M/C4G) turnover showed similar results where the odds of treatment discontinuation were significantly increased in patients with values above the given cut-off (OR [95% CI]: CPa9-HNE/C4G 11.33 [1.25–136.10]; C3M/PRO-C3 13.00 [2.12–54.97]; C4M/C4G 11.25 [2.08–58.41] *p* < 0.05 for all).

### 2.4. The Proportion of Long-Term VEDO Users Decreases in a Concentration-Dependent Manner across Tertile Levels of Collagen Turnover and Neutrophil Activity

To further examine the associations between the analyzed biomarkers and long-term VEDO treatment, biomarker concentrations were divided into tertiles. Tertile ranges with their corresponding proportion of long-term VEDO users for all biomarkers measured are shown in Appendix A. The proportion of long-term VEDO users was highest in the first tertile for biomarkers of type I (C1M) and III (C3M) collagen degradation, and the ratio between type III collagen formation and degradation (C3M/PRO-C3). (C1M: 91% vs. 45% and 20%, *p* = 0.004; C3M: 82% vs. 45% and 30%, *p* = 0.049; C3M/PRO-C3: 82% vs. 45% and 30%, *p* = 0.049). The proportion of long-term users decreased in a concentration-dependent manner across the tertiles, indicating that patients with the lowest concentrations of these biomarkers less frequently discontinued treatment within the first 12 months (Figure 3A–C). For biomarkers of basement membrane turnover (PRO-C4), collagen IV turnover (C4M/C4G), and neutrophil activity (CPa9-HNE), the majority of long-term users were observed within the first and second tertiles, as opposed to the third tertile (PRO-C4: 73% and 64% vs. 20%, *p* = 0.037; C4M/C4G: 73% and 64% vs. 20%, *p* = 0.037; CPa9-HNE: 82% and 55% vs. 20%, *p* = 0.018) (Figure 3D–F). Of note, the third tertile of PRO-C3 had the highest percentage of long-term users for all the biomarkers, or 70%. Although not significant, a trend was observed within biomarkers for type IV (C4M) and VI (C6Ma3) collagen degradation, where most long-term users were ranked within the first tertile when compared with the second and third tertile (C4M: 82% vs. 36% and 40%, *p* = 0.062; C6Ma3: 82% vs. 36% and 40%, *p* = 0.062) (Appendix A). Proportions of primary (PR) and secondary responders (SR) within each tertile are also presented in Appendix A. 

## 3. Discussion

In this study, we demonstrate that serological biomarkers of neutrophil activity and collagen turnover are associated with long-term use of vedolizumab treatment in patients with CD. We observed that patients who discontinued VEDO therapy within 12 months had elevated baseline levels of MMP-mediated type I, III, IV, and VI collagen degradation. Baseline levels of basement membrane turnover, the ratio between type III collagen formation and degradation, and two distinct type IV collagen turnover ratios were also significantly higher in patients who discontinued VEDO treatment than in patients who continued the treatment. Moreover, patients who discontinued VEDO treatment had significantly elevated levels of the neutrophil activity marker CPa9-HNE compared with long-term users. Synchronously, we observed that the proportion of long-term VEDO users decreased in a concentration-dependent manner across tertile levels of the biomarkers.

Firstly, it is noteworthy to mention the unique position of the study cohort regarding prior anti-TNF exposure, as the majority of included patients had previously experienced anti-TNF failure due to loss of response or adverse events. Several studies have been conducted on phenotypic disease differences between anti-TNF-naïve vs. anti-TNF-exposed patients with CD. For instance, Verstockt et al. observed a negative association between previous anti-TNF exposure and endoscopic remission after vedolizumab therapy, where anti-TNF-naïve patients had significantly better outcome rates [17]. Evidence also points to previous anti-TNF exposure being associated with decreased efficiency in preventing postoperative disease recurrences [18]. In a study investigating the contribution of intestinal fibrosis to incomplete response to anti-TNF, they observed that anti-TNF-exposed patients displayed increased deposition of type I and III collagen in the mucosa compared with anti-TNF-naïve patients. Both groups had a comparable histological inflammation score, leaving subclinical phenotypic features such as microscopic fibrosis as the culprit for insufficient response to anti-TNF [19]. Taken together, these data suggest that patients previously exposed to anti-TNF have a more severe disease manifestation when compared with anti-TNF-naïve patients. This highlights the importance of early therapeutic decision-making.

The biomarkers of ECM turnover applied in this study are derived from post-translational modifications of collagens by proteases such as MMPs, developed using the Protein FingerPrint™ technology [20]. The intestinal ECM is a collagen-rich matrix, and imbalanced ECM remodeling equilibrium along with increased protease activity are hallmarks of IBD. Therefore, these biomarkers served as potential surrogate measures of mucosal destruction and a disrupted intestinal barrier with regard to treatment response, since they reflect important layers of the intestinal tissue. Type I and III collagen are the main collagens of the interstitial matrix (IM), while type IV collagen is the main constituent of the basement membrane (BM). Type VI collagen can be found in the interface between the BM and IM. To the best of our knowledge, however, there are no integrative ECM remodeling biomarkers available that reflect molecular processes and changes in the intestinal tissue simultaneously.

The present study demonstrates that patients who discontinued VEDO therapy within the first 12 months had elevated baseline levels of MMP-mediated type I, III, IV, and VI collagen degradation compared with patients who continued VEDO therapy for at least 12 months. Studies have shown that MMPs are abundantly expressed in IBD by various immune cells in response to chronic inflammation [11]. Results from immunohistochemical expression in IBD showed that macrophages present in a disrupted intestinal microenvironment were positive for MMP-12 expression, while fibroblast-like cells were a source of MMP-13 [21]. Additionally, MMP-9 has been demonstrated to be highly produced by mucosal myofibroblasts, while neutrophils mainly produce MMP-9 [22]. Furthermore, it has been shown that MMP-9 is higher in sera from patients with IBD than in healthy controls and in patients with active CD than inactive disease [23]. Considering these findings, the collagen degradation biomarkers may be linked to chronic inflammation and disease activity in IBD.

In our study, patients who discontinued VEDO treatment had elevated degradation levels already at baseline before treatment initiation, suggesting excessive ECM remodeling due to severe disease. Notably, these elevated biomarkers reflect all layers of the intestinal tissue: type IV collagen, measured by C4M, is the most abundant collagen of the BM, mirroring epithelial integrity; type I and III collagen, measured by C1M and C3M, are the major collagens of the IM and represent the deeper tissue layers; type VI collagen measured by C6Ma3 is highly expressed at the interface between the BM and IM and is essential for maintaining tissue stability. This supports the idea that compared with long-term VEDO users, patients who discontinued VEDO treatment present with greater mucosal damage and loss of tissue integrity before treatment initiation, which may lessen their likelihood of achieving treatment response. In addition, we observed that patients who discontinued VEDO also had elevated baseline levels of BM turnover (PRO-C4), type III collagen turnover (C3M/PRO-C3), and two type IV collagen turnover ratios (C4M/C4G, PRO-C4/C4G). Similar results have been shown where degradation of collagen and BM turnover have been linked to the pathology of CD. A study conducted by Lindholm et al. showed that biomarkers of type III (C3M) and IV (C4M) collagen degradation, along with the BM turnover marker PRO-C4, were elevated in serum from rats with dextran sulfate sodium (DSS)-induced colitis. Additionally, the turnover of type III collagen (C3M/PRO-C3) demonstrated a positive correlation to the histological manifestation of the disease [24]. Another study involving serological biomarkers of type VI collagen turnover demonstrated that baseline levels of MMP-mediated type VI collagen degradation were significantly increased in CD patients with mild and moderate-to-severe endoscopically active disease compared with endoscopic remission [25].

The neutrophil activity marker CPa9-HNE was also significantly higher in patients who discontinued VEDO therapy compared with long-term users. The neo-epitope detected here is a HNE-derived fragment of calprotectin, a protein that comprises about 45% of neutrophil cytosolic proteins [26]. It is involved in the recruitment of leukocytes and has antimicrobial activity toward pathogens [27]. Increased fecal calprotectin levels are a known biochemical indicator of intestinal inflammation and it is currently the most widely used biomarker of inflammation in IBD [28]. Neutrophils are the most prevalent immune cells in the body and are known for their rapid recruitment to sites of inflammation. When activated, they recruit and activate a broad spectrum of other immune cells and cause tissue damage by releasing proteases such as MMPs and HNE [29]. HNE is a multifunctional serine protease mainly involved in regulating inflammatory processes and killing pathogens. Its proteolytic activity is usually tightly controlled to avoid the exuberant breakdown of ECM proteins [30]. Plasma levels of HNE have shown to be higher in patients with active Crohn’s disease than in healthy individuals [31]. These data corroborates our previously discussed findings, where patients who discontinue VEDO therapy may have significant mucosal damage and persistent inflammation at baseline compared with long-term VEDO users.

Tertile biomarker levels of type I, III, IV collagen degradation, BM turnover, type III and IV collagen turnover, and serum calprotectin were associated with the frequency of long-term VEDO use. Most patients who became long-term users were ranked within the first tertile, having the lowest biomarker concentration, while the highest tertiles contained fewer long-term VEDO users. Biomarker levels divided into tertiles highlight that these markers could be utilized in clinical practice for optimal decision-making regarding the choice of treatment, i.e., physicians could decide only to treat patients who fall within the first two tertiles since patients within the third tertile would be less likely to benefit from the treatment.

Several study limitations warrant recognition. First, this study was of a retrospective nature, for which clinical outcome data were collected based on available reports, whereas no strict follow-up visits were conducted after 12 months of treatment initiation. This may have introduced information bias since we were forced to rely on others for accurate recordkeeping. Data from strict follow-up visits could have helped us to more accurately determine the predictive value of the biomarkers in relation to pre-defined response criteria instead of only treatment continuation. Second, we had to rely on a clinical and serological assessment of disease activity (i.e., HBI scores, PGA, and CRP), as data on fecal calprotectin levels or endoscopic disease activity were not sufficiently recorded at the time of sampling. Third, our sample size was limited. However, the main reason for this was that there are generally not many patients with CD receiving VEDO treatment at the UMCG, which limited the inclusion. Finally, we could not perform an external validation of our results since we did not have a suitable replication cohort available for this purpose. The strengths of this study include the unique nature of our cohort, given that cohorts of patients with CD starting vedolizumab treatment are relatively sparse in literature and rarely subjected to biomarker assessment. We could link the biomarkers to long-term treatment continuation, which is relevant considering that therapeutic effects of VEDO generally take more time to become clinically apparent compared with first-line biologicals such as TNF-α-antagonists, especially in anti-TNF-experienced patients [32]. Therefore, long-term assessment is highly preferred over early (induction) treatment effects. Prospective longitudinal studies are needed to clinically validate findings like these, since biomarker analyses in a retrospective setting are generally hypotheses-generating. Prospectively defining a study population also has other advantages such as a uniformly defined study cohort, and in terms of statistical analysis, it can limit power issues and eliminate confounding factors. Therefore, external validation in an independent prospective cohort of patients with CD should be performed to validate the predictive capability of the investigated biomarkers. If possible, disease activity and severity should be evaluated by modalities such as cross-sectional imaging.

## 4. Materials and Methods

### 4.1. Study Design and Population

In this retrospective observational cohort study, serum was collected from 33 patients with CD who initiated vedolizumab induction therapy at the IBD center of the University Medical Center Groningen (UMCG, Groningen, the Netherlands) in 2011–2018 (Figure 4). The diagnosis of CD was based on clinical, endoscopic, and histological criteria. Inclusion criteria were an established diagnosis of CD for at least one year, age of ≥18 years, and having an active disease based on a combination of clinical scores, biochemical measures (e.g., C-reactive protein (CRP), fecal calprotectin, or both), and endoscopic assessment, as an indication for VEDO therapy. Exclusion criteria were as follows: patients who underwent any type of surgery or endoscopic balloon dilatation <6 months before sampling, patients with concurrent malignancies (except for skin cancer and hematological malignancies), other fibrotic diseases (e.g., liver fibrosis/cirrhosis, lung fibrosis), concurrent infections, or perianal disease as the only indication for starting vedolizumab induction therapy. Blood samples were drawn just before induction at baseline. At the time of sampling, detailed phenotypic data were collected for all patients, such as the Montreal disease classification, medication use, history of intestinal surgery, disease activity, and standard laboratory parameters. This study was designed to compare serum biomarker levels of ECM formation and degradation between patients who continued with VEDO treatment for at least 12 months and those who did not. Long-term treatment continuation was used as a surrogate measure for response to treatment, as no strict follow-up visits were conducted at this time point. Reasons for discontinuation of VEDO treatment were loss of response, non-response, or severe adverse events, as evaluated by the treating gastroenterologist. Patients using VEDO > 12 months were further stratified into those who achieved primary or secondary response. Primary response was defined as steroid-free remission at week 14 (Harvey–Bradshaw Index (HBI) < 5 and/or by physician’s global assessment (PGA)) and continuation of VEDO for at least 12 months after the start of therapy. Secondary response was defined as non-response at week 14 (HBI ≥ 5 and/or non-response as determined by PGA), but with continuation of VEDO for at least 12 months after the start of therapy. This study was approved by the Institutional Review Board (IRB) of the UMCG (IRB no. 08/338). All patients provided written informed consent for the use of patient data and serum. The study was conducted according to the principles of the Declaration of Helsinki (2013).

### 4.2. Biomarker Assays

Serological biomarkers of neo-epitope fragments reflecting ECM degradation or formation were measured using a solid-phase competitive enzyme-linked immunosorbent assay (ELISA). The markers included in this study and references to their technical papers, where information on validation parameters and detailed assay conditions can be found, are listed in Table 5. Assays were either colorimetric-based or chemiluminescence-based. Ninety-six (96)-well plates pre-coated with streptavidin were coated with 100 µL of biotinylated antigen corresponding to the fragment of interest for 30 min at 20 °C while shaking at 300 rpm. Each incubation step was followed by washing plates with washing buffer (25 mM TRIZMA, 50 mM NaCl, 0.036% Bronidox L4, 0.1% Tween-20). Standard curves were serially diluted in assay buffer and added to the wells, along with serum samples diluted in assay buffer, and assay controls. Subsequently, 100 µL of fragment-specific horseradish peroxidase-conjugated monoclonal antibody was added to the wells and incubated for either 1 h at 20 °C or 20 h at 4 °C while shaking at 300 rpm. For colorimetric-based assays, tetramethylbenzidine (TMB) (Kem-En-Tec, CAT no. 438OH, Taastrup, Denmark) was added (100 µL/well) and incubated for 15 min in darkness at 20 °C. Stop buffer (1% H_2_SO_4_) was added to the wells (100 µL/well) to stop the reaction. The absorbances were read at 450 with 650 nm as a reference, using a VersaMAX ELISA microplate reader (Molecular Devices, Wokingham Berkshire, UK). For chemiluminescence-based assays, 100 µL of BM Chemiluminescence ELISA Substrate (Merck, Denmark) was added to the wells and shaken for 1 min at 300 rpm before incubating for 2 min. Plates were read using a fluorescence plate reader (Fluroskan, FL, Thermo Fisher, Denmark), with a read light emission at 1000 ms and no filter. Standard curves were plotted using a four-parameter logistic (4PL) model.

### 4.3. Statistical Analysis

All data were treated non-parametrically after visual assessment using density plots. Differences in baseline characteristics between responders and non-responders were compared using Mann–Whitney *U*-tests for continuous variables, presented as medians with interquartile ranges (IQRs), and Fisher’s exact tests for categorical variables, presented as proportions *n* with corresponding percentages (%). Serum biomarker levels were presented as medians [IQR]. Differences between groups were compared using Mann–Whitney *U*-tests, or Kruskal–Wallis tests followed by Dunn tests with post-hoc Holm correction for multiple comparisons. Spearman’s rank correlation coefficients were calculated to determine associations between baseline biomarker levels and disease activity scores (baseline HBI and SES-CD < 3 months from baseline). Furthermore, a binary variable was created based on HBI at baseline, reporting whether patients had an active (HBI > 5) or inactive (HBI < 5) disease. To assess the predictive capability of the biomarkers with regards to differentiating between long-term users of VEDO and patients who discontinued treatment within 12 months, area under the curve (AUC) values were computed as an evaluation metric using receiver operating characteristics (ROC) statistics. To explore the predictive value of biomarker panels, biomarkers were empirically combined in a multivariable logistic regression model (method: enter), and AUC values were obtained. All AUC values were presented with their corresponding 95% confidence intervals. Sensitivity and specificity metrics were obtained by determining the optimal cut-off using Youden‘s *J* statistic. Odds ratios were calculated for each specified cut-off value, and *p*-values were obtained using Fisher‘s exact tests. Biomarker levels were divided into tertiles, and chi-square tests were used to investigate potential differences in biomarker tertiles and proportions of long-term users/non-users. *p*-values ≤ 0.05 were considered statistically significant. Statistical analysis was performed using R version 4.1.2 (Rstudio, Boston, MA, USA). Figures were designed using GraphPad Prism version 9.2.0 (La Jolla, CA, USA).

## 5. Conclusions

In conclusion, this study demonstrates that serological biomarkers of neutrophil activity and type I, III, IV, and VI collagen turnover are elevated at baseline in patients who discontinued VEDO therapy within the first 12 months compared with long-term VEDO users. These biomarkers are linked to the pathology of CD, have predictive capabilities, and could therefore be used for early decision-making concerning VEDO therapy. Future studies are warranted to validate these biomarkers and assess their predictive value for guiding personalized treatment decisions.

## Figures and Tables

**Figure 1 ijms-23-08137-f001:**
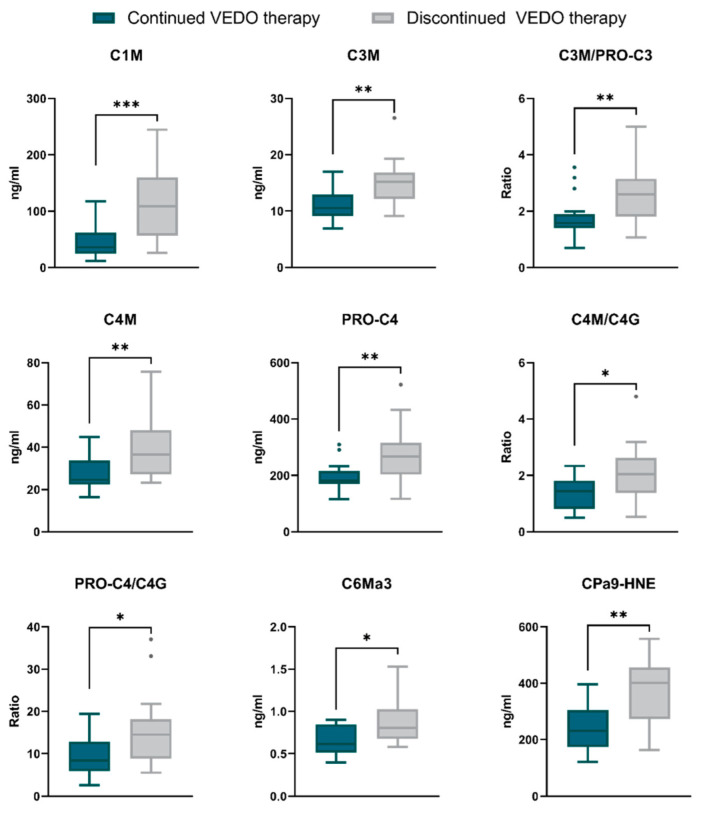
Baseline serum biomarker levels between patients with long-term continuation and discontinuation of VEDO treatment. Biomarker levels are presented as Tukey boxplots. * *p* < 0.05; ** *p* < 0.01; *** *p* < 0.001.

**Figure 2 ijms-23-08137-f002:**
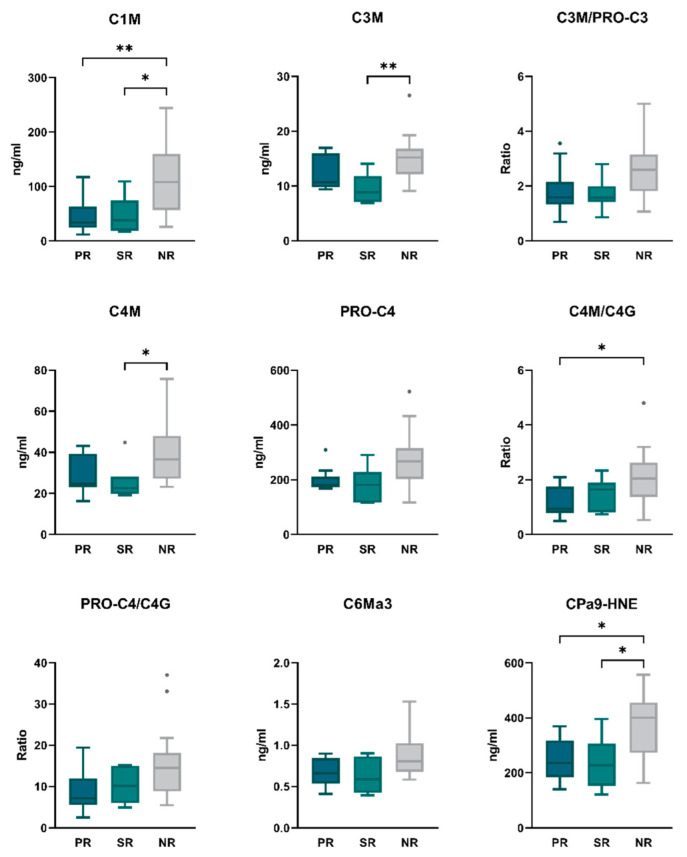
Baseline serum biomarker levels between primary responders (PR), secondary responders (SR), and non-responders (NR) to VEDO treatment. Biomarker levels are presented as Tukey boxplots. * *p* < 0.05; ** *p* < 0.01.

**Figure 3 ijms-23-08137-f003:**
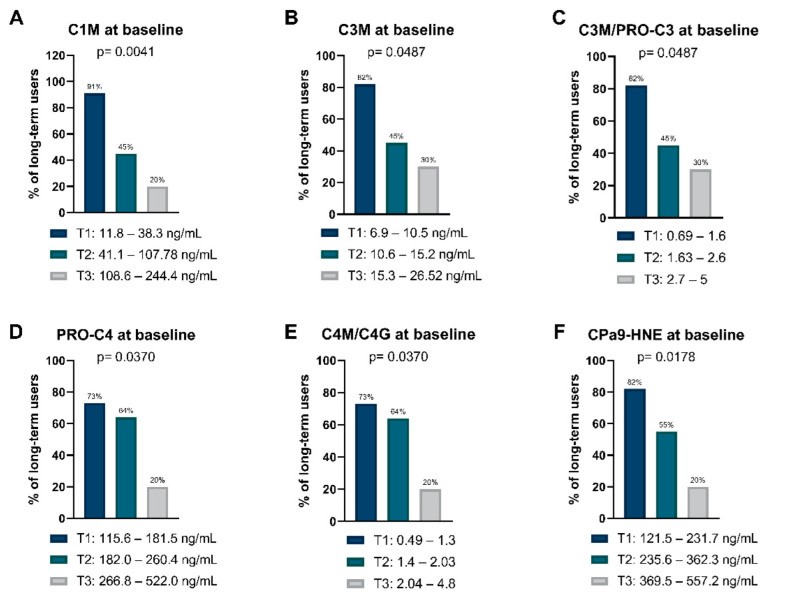
(**A**–**E**) Tertile biomarker levels of type I, III, and IV collagen degradation, BM turnover, type III and IV collagen turnover, as well as (**F**) serum calprotectin. Bars represent the percentage (%) of patients who became long-term users within each tertile, with their corresponding ranges listed below.

**Figure 4 ijms-23-08137-f004:**
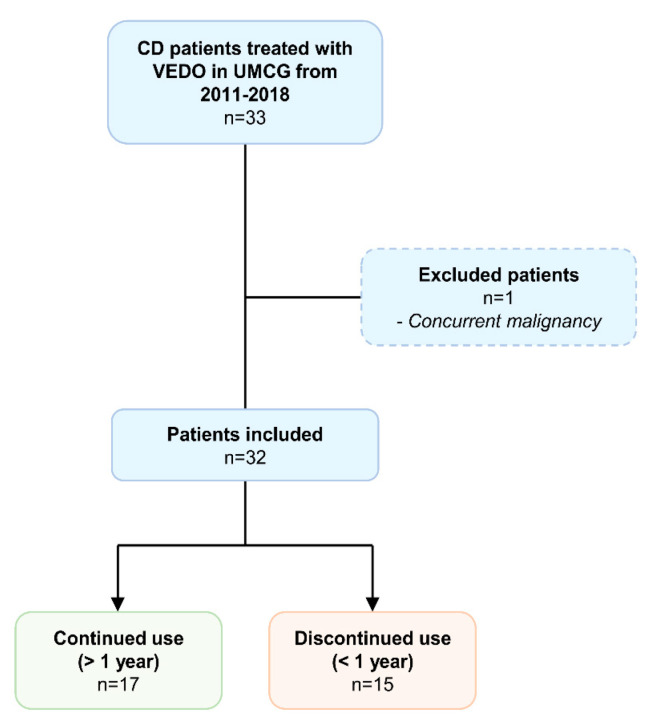
Patient inclusion flowchart.

**Table 1 ijms-23-08137-t001:** Patient baseline characteristics and clinical demographics, stratified by long-term use of VEDO treatment as a surrogate outcome for sustained response to treatment.

	Continued Use (*n* = 17)	Discontinued Use (*n* = 15)	*p*
**Age (years)**	41 [35–60]	42 [35–60]	>0.999
**Gender**			0.444
Female	6 (35%)	3 (20%)	
Male	11 (65%)	12 (80%)	
**BMI (kg/m^2^)**	22.6 [21.0–25.0]	24.2 [20.9–26.9]	0.509
**Smoking**			0.134
No	5 (29%)	10 (67%)	
Previous	4 (24%)	2 (13%)	
Current	8 (47%)	3 (20%)	
**Montreal classification**			
**Montreal Age (A)**			0.886
A1 (≤16 years)	2 (12%)	3 (20%)	
A2 (17–40 years)	11 (65%)	8 (53%)	
A3 (>40 years)	4 (24%)	4 (27%)	
**Montreal Location (L)**			0.501
L1 (ileal disease)	4 (24%)	2 (13%)	
L2 (colonic disease)	1 (5.9%)	3 (20%)	
L3 (ileocolonic)	12 (71%)	10 (67%)	
L4 (upper GI disease)	0 (0%)	0 (0%)	
**Montreal Behavior (B)**			0.553
B1 (non-stricturing, non-penetrating)	5 (29%)	5 (33%)	
B2 (stricturing)	8 (47%)	4 (27%)	
B3 (penetrating)	4 (24%)	6 (40%)	
**Montreal Perianal disease (*p*)**	5 (29%)	4 (27%)	>0.999
**Medication use, *n* (%)**			
Aminosalicylates	0 (0%)	2 (13%)	0.212
Steroids	9 (53%)	9 (60%)	0.688
Immunosuppressives	9 (53%)	4 (27%)	0.131
Prior anti-TNF-α	15 (88%)	15 (100%)	0.486
Prior Vedolizumab	0 (0%)	(0%)	NA
**Surgical history**			
Colectomy	0 (0%)	1 (6.7%)	0.469
Ileocecal resection	11 (65%)	9 (60%)	0.784
**Clinical disease activity**			
**Harvey-Bradshaw Index (HBI)**			0.866
Remission (<5)	1 (7.1%)	2 (17%)	
Mild disease (5–7)	6 (43%)	4 (33%)	
Moderate disease (8–16)	7 (50%)	6 (50%)	
Severe disease (>16)	0 (0%)	0 (%)	
**Clinical parameters**			
Hemoglobin (nmol/L)	7.60 [7.00–8.20]	7.50 [7.20–7.95]	0.583
WBC (×10^9^/L)	7.3 [6.4–9.1]	9.4 [5.0–12.3]	0.821
Neutrophil count (×10^9^/L)	4.91 [3.96–5.83]	5.54 [3.17–8.98]	0.796
Eosinophil count (×10^9^/L)	0.16 [0.03–0.24]	0.09 [0.05–0.17]	0.404
CRP (mg/L)	2 [1–6]	14 [8–26]	0.001
Creatinine (µmol/L)	61 [49–71]	67 [59–72]	0.508
eGFR (mL/min/1.73 m^2^)	108 [94–120]	104 (82–122]	0.664
Fecal calprotectin (µg/g) ^^^	982 [658–1455]	1400 [852–1825]	0.475

Data are presented as median [IQR] for continuous variables or as proportions *n* with corresponding percentages (%) for categorical variables. ^ Fecal calprotectin levels were available for *n* = patients.

**Table 2 ijms-23-08137-t002:** Baseline serum biomarker levels stratified by long-term response to vedolizumab. Values are displayed as medians with IQR.

Biomarker	Continued Use *n* = 17	Discontinued Use *n* = 15	*p*-Value
**C1M (ng/mL)**	**36.1 [25.02–49.80]**	**108.6 [57.47–148.45]**	**0.001**
**C3M (ng/mL)**	**10.5 [9.43–11.81]**	**15.2 [13.07–16.44]**	**0.006**
PRO-C3 (ng/mL)	6.4 [4.43–8.40]	5.7 [5.34–6.49]	0.610
**C3M/PRO-C3**	**1.6 [1.42–1.80]**	**2.6 [1.98–3.13]**	**0.008**
**C4M (ng/mL)**	**24.5 [22.36–28.16]**	**36.5 [28.24–46.68]**	**0.010**
C4G (ng/mL)	19.4 [15.19–30.24]	18.0 [13.56–26.74]	0.558
**PRO-C4 (ng/mL)**	**182.0 [170.48–203.68]**	**266.8 [207.20–308.26]**	**0.010**
**C4M/C4G**	**1.4 [0.81–1.77]**	**2.0 [1.40–2.44]**	**0.020**
PRO-C4/C4M	7.2 [6.10–8.06]	6.9 [6.51–7.75]	0.925
**PRO-C4/C4G**	**8.4 [6.02–12.56]**	**14.5 [9.66–17.12]**	**0.033**
**C6Ma3 (ng/mL)**	**0.6 [0.53–0.84]**	**0.8 [0.68–1.02]**	**0.015**
**CPa9-HNE (ng/mL)**	**231.7 [195.32–302.60]**	**401.2 [281.36–452.00]**	**0.003**
CPa9-HNE/C4G	11.6 [6.46–16.43]	18.5 [11.14–29.28]	0.052

**Table 3 ijms-23-08137-t003:** Receiver operating characteristics (ROC) analysis demonstrating the area under the curve (AUC) of each individual biomarker as well as a selected combination of biomarkers, representing their discriminative ability regarding long-term continuation vs. discontinuation of vedolizumab treatment.

Biomarker	AUC [95% CI]	Sensitivity (%)	Specificity (%)	*p*-Value
**Individual biomarkers**				
**C1M**	**0.85 [0.72–0.98]**	**76**	**87**	**<0.001**
**CPa9-HNE**	**0.81 [0.66–0.96]**	**100**	**53**	**<0.001**
**C3M**	**0.79 [0.62–0.95]**	**71**	**87**	**0.001**
**C3M/PRO-C3**	**0.78 [0.60–0.95]**	**76**	**80**	**0.002**
**C4M**	**0.77 [0.60–0.93]**	**76**	**73**	**0.002**
**PRO-C4**	**0.77 [0.59–0.94]**	**71**	**80**	**0.003**
**C6Ma3**	**0.75 [0.58–0.92]**	**53**	**93**	**0.004**
**C4M/C4G**	**0.74 [0.56–0.92]**	**88**	**60**	**0.009**
**PRO-C4/C4G**	**0.72 [0.54–0.90]**	**82**	**53**	**0.015**
CPa9-HNE/C4G	0.70 [0.52–0.89]	100	40	0.034
PRO-C4/C4M	0.49 [0.28–0.70]	100	13	0.927
PRO-C3	0.45 [0.24–0.66]	35	87	0.623
C4G	0.44 [0.23–0.65]	35	67	0.563
**Combined biomarkers**				
**[PRO-C4, C6Ma3]**	**0.84 [0.70–0.98]**	**65**	**93**	**<0.001**
**[C3M, C6Ma3]**	**0.83 [0.69–0.97]**	**71**	**87**	**<0.001**
**[PRO-C4, C3M/PRO-C3]**	**0.81 [0.65–0.97]**	**100**	**53**	**<0.001**

Abbreviations: AUC, area under the curve; CI, confidence interval.

**Table 4 ijms-23-08137-t004:** Odds ratios (ORs) for the optimal cut-off values, determined by Youden’s J statistics on the respective ROC curves.

Biomarker	OR [95% CI]	*p*-Value
**C1M**	**14.08 [2.35–59.09]**	**0.002**
**C3M**	**15.60 [2.79–80.36]**	**0.002**
PRO-C3	3.56 [0.61–19.27]	0.229
**C3M/PRO-C3 ^a^**	**13.00 [2.12–54.97]**	**0.004**
**C4M**	**8.93 [1.62–42.45]**	**0.012**
C4G	1.09 [0.2677–4.960]	>0.999
**PRO-C4**	**9.60 [1.67–39.52]**	**0.006**
**C4M/C4G ^a^**	**11.25 [2.08–58.41]**	**0.008**
PRO-C4/C4M ^a^	5.33 [1.19–21.94]	0.062
PRO-C4/C4G ^a^	2.62 [0.27–39.81]	0.579
**C6Ma3**	**15.75 [2.01–182.70]**	**0.007**
**CPa9-HNE**	**19.43 [2.36–255.20]**	**0.004**
**CPa9-HNE/C4G ^a^**	**11.33 [1.25–136.10]**	**0.030**

Abbreviations: OR, odds ratios; CI, confidence interval. Cut-off values based on specificity and sensitivity are shown in table. ^a^ Biomarker ratios with no concentration unit.

**Table 5 ijms-23-08137-t005:** Serological biomarkers of extracellular matrix formation/degradation, intestinal inflammation, and immune cell activity measured in this study.

Protein	Biomarker of Degradation	Biomarker of Formation	Implication	References
Type I collagen	**C1M**: Neo-epitope of MMP-2, -9, -13 mediated degradation of type I collagen	-	IM degradation	[33]
Type III collagen	**C3M**: Neo-epitope of MMP-9 mediated degradation of type III collagen	**PRO-C3**: Released N-terminal pro-peptide of type III collagen	IM turnover	[34,35]
Type IV collagen	**C4M**: Neo-epitope of MMP-2, -9, -12 mediated degradation of type IV collagen alpha-1 chain **C4G**: Neo-epitope generated by T-cell granzyme-B-mediated degradation of type IV collagen	**PRO-C4**: Internal epitope in 7s domain of type IV collagen	BM turnover	[36,37,38]
Type VI collagen	**C6Ma3**: MMP-2 and -9 degraded type VI collagen	-	BM/IM degradation	[39]
Calprotectin	**CPa9-HNE**: Neo-epitope of human neutrophil elastase (HNE) mediated degradation of calprotectin	-	Neutrophil activity	[40]

## Data Availability

The data presented in this study are available upon reasonable request from the corresponding author.

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
