# Peer review of "Serological Biomarkers of Extracellular Matrix Turnover and Neutrophil Activity Are Associated with Long-Term Use of Vedolizumab in Patients with Crohn’s Disease"

_ijms, 2022, doi:10.3390/ijms23158137_

Round 1

Reviewer 1 Report

I would change the order of the different sections since the material and methods are confusing in the last place.

It would be good to add the importance of prospective studies in this field.

Author Response

Dear reviewer,

We are excited for the opportunity to submit a revised version of our manuscript “Serological biomarkers of extracellular matrix turnover and neutrophil activity are associated with long-term use of vedolizumab in patients with Crohn’s disease” for publication in International Journal of Molecular Sciences.

We would like to thank you and the editorial team for the enthusiastic support of our work and we welcome all constructive comments, further allowing us to improve the quality of our manuscript. We appreciate the opportunity to address the comments and have done so in a point-by-point manner below. Changes made in the revised manuscript are tracked and line references (“lines xxx-xxx”) refer to the revised manuscript file.

  1. “I would change the order of the different sections since the material and methods are confusing in the last place.”
  • Authors’ reply: We agree with the reviewer that normally the Material and Methods would be placed before Results. However, the reason for this arrangement is that according to the journal template used, and the instructions for authors (Research Manuscript Sections), Materials and Methods should be placed after Discussion, before Conclusions.

  1. “It would be good to add the importance of prospective studies in this field.”
  • Authors’ reply: The reviewer brings up an appropriate point. It is indeed important to study and validate the clinical application of biomarkers such as these in prospective longitudinal studies. In the Discussion section, we have devoted a few sentences addressing this (lines 380-387).
  • Prospective longitudinal studies are needed to clinically validate findings like these, since biomarker analyses in a retrospective setting are generally hypotheses-generating. Prospectively defining a study population also has other advantages such as a uniformly defined study cohort, and in terms of statistical analysis, it can limit power issues and eliminate confounding factors. Therefore, external validation in an independent prospective cohort of patients with CD should be performed to validate the predictive capability of the investigated biomarkers. If possible, disease activity and severity should be evaluated by modalities such as cross-sectional imaging.

Reviewer 2 Report

1. The section Materials and Methods should be written after Introduction and before Results.

2. In the Discussion chapter you should add a paragraph about other serological biomarkers of extracellular matrix turnover and further to explain why you choose matrix metalloproteinase (MMP)-derived fragments of type I (C1M), III (C3M), IV 20 (C4M), and VI (C6Ma3) collagen, type III collagen formation (PRO-C3), basement membrane turn- 21 over (PRO-C4) and T-cell activity (C4G).

3. You should add more information about Future perspectives
